# Effects of Red-Bean Tempeh with Various Strains of *Rhizopus* on GABA Content and Cortisol Level in Zebrafish

**DOI:** 10.3390/microorganisms8091330

**Published:** 2020-08-31

**Authors:** Yo-Chia Chen, Shu-Ling Hsieh, Chun-Yi Hu

**Affiliations:** 1Department of Biological Science and Technology, National Pingtung University of Science and Technology, Pingtung 912301, Taiwan; ox@mail.npust.edu.tw; 2Department of Seafood Science, National Kaohsiung University of Science and Technology, Kaohsiung 81157, Taiwan; slhsieh@webmail.nkmu.edu.tw; 3Department of Food Science and Nutrition, Meiho University, Pingtung 912009, Taiwan

**Keywords:** tempeh, γ-aminobutyric acid (GABA), antibacterial activity, anti-stress, cortisol

## Abstract

Tempeh is traditionally produced by fermenting soybean with the fungus *Rhizopus oligosporus* found in banana leafs. We wanted to investigate if Taiwan’s flavorful red bean could be used as a healthy substitute for soybeans in tempeh. One bioactive component of tempeh is *γ-Aminobutyric* acid (GABA). We measured GABA content and shelf-life-related antimicrobial activity in red-bean tempeh made with four strains of *Rhizopus*, one purchased strain of *Rhizopus*, and an experimental co-cultured group (*Rhizopus* and *Lactobacillus rhamnosus* BCRC16000) as well as cortisol in red-bean-tempeh-treated zebrafish. GABA was highest in the co-culture group (19.028 ± 1.831 g kg^−1^), followed by screened Strain 1, the purchased strain, and screened Strain 4. All strains had antibacterial activity on *S. aureus* and *B. cereus*. The extract significantly reduced cortisol in zebrafish. However, Strain 1, with less GABA than some of the other strains, had the best effect on cortisol level, suggesting that other components in red-bean tempeh may also affect stress-related cortisol. We found the benefits of red-bean tempeh to be similar to those reported for soybean-produced tempeh, suggesting that it could be produced as an alternative product. Considering the Taiwanese appreciation of the red-bean flavor, it might find a welcoming market.

## 1. Introduction

The red bean (*Phaseolus vulgaris*) is an herbaceous annual plant of the family leguminosae [1]. Beans in general are important sources of macronutrients, micronutrients and antioxidant compounds and can hold great potential to improve human and animal nutrition [2]. Plant foods are fermented to enhance or create unique flavors, to change the textural properties, and to improve quality and digestibility. Fermented foods are an essential part of the human diet in many parts of the world, especially in Southeast Asia, the Near East, and parts of Africa [3].

*Rhizopus oligosporus* has been used in Indonesia to produce soybean tempeh (tempe kedele) since ancient times, and interest in this food has increased worldwide [4]. It is normally fried, boiled, steamed or roasted. During incubation with *R. oligosporus*, the soybeans are bound together by the white mycelium, forming a cake, and enzymes released by the fungus render the protein-rich product more digestible to humans [5]. Legumes other than soybeans, in addition to cereals and process by-products, can also be used as substrates for tempeh [6].

The fermentation process of tempeh increases the nutritional values of some nutrients, and the development of vitamins, phytochemicals and antioxidative constituents [7]. Isoflavone levels in tempeh are relatively high as compared with other soybean products such as tofu and soy beverages [8]. Other benefits of the fermentation process of tempeh are a decrease in the level of phytic acid and enhancement of the bioavailabilities of minerals, such as calcium, zinc and iron [7].

A variety of microorganisms, including filamentous fungi, yeasts and bacteria, are found in traditional, commercial and laboratory-made soybean tempeh [9]. Lactic acid bacteria (LAB), in particular, contribute to the fermentation process and ensure the safety of soybean tempeh [10]. However, different LAB species are adapted to different environments and substrates [11], and fungi–bacteria interactions can impact the course of fungal infection and biotechnological use.

Ross et al. found that fungus positively influences the growth of bacteria and dramatically increases bongkrekic acid production in stationary culture, which inhibits fungus growth. These results demonstrate the context-dependent formation of antifungal and antibacterial agents at the fungal–bacterial interface, which may also serve as a model for scenarios observed in mixed infections [12].

Little is known about *Rhizopus* species with regard to their cell-wall glycoconjugates as well as their specific functions in the fungal cell and their interactions with the environment. Ceramide monohexosides (CMHs) are highly-conserved fungal glycosphingolipids with structural modifications that include different sites of unsaturation as well as fatty-acid residues of varying lengths in their ceramide moieties [13]. CMHs play diverse roles in fungal cell processes, including growth and morphological transition in *Cryptococcus neoformans*, *Pseudallescheria boydii*, *Candida albicans*, *Aspergillus fumigatus* and *Collectotrichum gloeosporioides* [14,15]. In addition, CMHs are important in promoting alkaline tolerance in vitro due to their involvement in the regulation of membrane fungal lipid domains, which leads to the redistribution of CMHs in the membrane in an alkaline environment [16]. CMHs also interact with defensins isolated from insects and plants [17].

γ-Aminobutyric acid (GABA), with a four-carbon non-protein amino acid structure, acts as the major inhibitory neurotransmitter of the central nervous system. Other physiological functions of GABA include the prevention of the induction of hypertension and diabetes, and diuretic and tranquilizer effects [18]. As such, GABA is used extensively in pharmaceutical preparations and functional foods such as dairy products, tea and alcohol [19]. Glutamate decarboxylase (GAD) is the enzyme that catalyzes the conversion of L-glutamate to GABA through a single-stepα-decarboxylation [20]. GABA works effectively as a natural relaxant and its effects can be seen within one hour of its administration to induce relaxation and diminish anxiety. Moreover, GABA administration can enhance immunity under stress [21].

The stress response system helps individuals deal with adverse conditions; for instance, during stress, an increased cortisol level can lead to hyperglycemia, providing energy for defensive actions [22]. Zebrafish (*Danio rerio*) can be used as the experimental model to test this possibility. This fish species has many advantages as a model organism, including easy handling, simple maintenance, and genetic homology with humans. For instance, environmental diazepam and fluoxetine have been found to interfere the stress response of the fish [23].

This study aimed to evaluate one important nutritional component and health effects of a tempeh we created using Taiwan red beans fermented with four strains of *Rhizopus* isolated from banana leaf. We measured the GABA content of the tempeh extracts and evaluated their antibacterial activity and their effect on cortisol levels in animals in a stress test.

## 2. Materials and Methods

### 2.1. Microorganisms

*R. microsporus* var. *oligosporus* IFO 8631 Reverse Osmosis (RO) (ATCC 22959) fungus strains and *Lactobacillus rhamnosus* BCRC 16000, *Escherichia coli* BCRC 10239, *Staphylococcus aureus* BCRC 10451, and *Bacillus cereus* BCRC 10927 bacteria were obtained from the Bioresource Collection Research Center (BCRC), Food Industry Research and Development Institute (Hsinchu, Taiwan).

For *Rhizopus* activation, 45 g of malt extract powder (Oxoid Ltd., Basingstoke, UK) was mixed with one liter of RO (Reverse Osmosis) water, heated, and stirred until boiling for one minute to completely dissolve the powder. Following sterilization at 120 °C for 20 min, the mixture was poured into a Petri dish and test tube, and cultured in Potato Dextrose Agar (Lab M Ltd., Heywood, Bury, UK) slant medium at 5 °C for five to seven days in order to observe the growth by microscope. The supernatant of the spores of the mycelium (approximately 1 × 10^6^/mL) was filtered through four layers of sterilized gauze (Belia, Jiangxi province, China) and the spore yield was calculated.

For bacterial cultivation, de Man, Rogosa, and Sharpe (MRS) medium [24] broth (Becton, Dickinson and Company, BD Life Science, Franklin Lakes, NJ, USA) was used as the storage medium. A total of 10 mL of bacterial solution was loaded into a spiral tube (10 × 120 mm), sterilized for fifteen minutes at 121 °C, 15 psi by autoclave (EZmed, EZ Media CO., Ltd., New Taipei City, Taiwan), and covered with 1.5 mL of degraded mineral oil (Paraffin oil, Tedia CO., Fairfield, CT, USA) at 90 °C to maintain a total anaerobic environment. The bacteria were cultivated in a 37 °C water bath overnight (approximately 12 h) and stored in a refrigerator at 4 °C. A total of 0.5 mL was taken before the experiment, inoculated into fresh MRS broth, and cultured in a 37 °C water bath for 8 h to obtain activated cells. In total, 1.11 mL of the bacterial solution was then inoculated into a dilution bottle containing 110 mL of fresh MRS broth. Finally, the dilution bottle was placed in an oscillating incubator and incubated at 37 °C, 120 rpm. In order to maintain the entire process anaerobically, a 5 mL injection syringe was used in the inoculation process.

### 2.2. Red Bean Fermentation

Red beans were washed and soaked for twelve hrs. After drying, water of twice the weight of red beans and 1% lactic acid were added, and followed by cooking at 100 °C for 30 min, and the cooked beans were then dried and cooled to 37 °C. A total of 100 g of cooked red beans was placed on an aluminum plate (10 × 20 cm) with an aluminum cover and thoroughly mixed with 5 mL of *R. microsporus* var. *oligosporus* IFO 8631 (ATCC 22959) containing 1 × 10^6^/mL spores. The mixture was incubated for two days in an aerobic environment at 37 °C. The fermented tempeh product was then freeze-dried for two days and ground to a powder for analysis. The above experiment was repeated at 30 °C. Samples were prepared according to a previously-described method with some modifications [25]. Briefly, one gram of fermented red beans was extracted using 15 mL of 70% alcohol and shaken for 1 min and put into 4 °C cold room for one hour and then centrifuged at 6000× *g* for 20 min. The whole extracting procedure was repeated three times before the supernatant was extracted and concentrated using a rotary evaporator (IKA RV10, Staufen im Breisgau, Germany) to 15 mL and stored in refrigerator.

### 2.3. Isolation of Rhizopus

One gram of banana leaf was mixed with 9 mL of RO water in a centrifuge tube and vortexed. The spread plate method was employed to culture on MEA with 1.5% agar (*w*/*v*) medium. After incubation for 24 h, different colonies were selected for culture using the streak plate method. After a further 24 h of incubation, the colonies were observed using a phase-contrast microscope (Eclipase 80i; Nikon, Japan), and the fungus was further purified by streak plate method with MEA + ampicillin (0339 Amresco, Solon, OH, USA).

### 2.4. DNA Isolation, Polymerase Chain Reaction (PCR) Amplification, and Sequencing

Strains were cultured for one day. Their mycelia were used as the DNA source. DNA was extracted using a Phytopure TM DNA Extraction Kit (GE Healthcare, Chicago, IL, USA) according to the manufacturer’s directions. Amplification of the ITS (Internal Transcribed Spacer) rDNA region was performed using PCR with the primer pair ITS1 (5′-TCCGTAGGTGAACCTGCGG-3′ and ITS4 (5′-TCCTCCGCTTATTGATATGC-3′) [26]. All PCR amplifications were performed in 25 mL of reaction mixture, containing ±100 ng of DNA template, 0.25 mM of each primer, PCR buffer 1, dNTPmix 0.2 mmol L^−1^, MgCl_2_ 1.75 mmol L^−1^, and 1 unit of Taq DNA polymerase. The reaction condition was set as follows: initial denaturation at 94 °C for 1 min, followed by 25 cycles of denaturation at 94 °C for 30 s, annealing at 55 °C for 30 s, and extension at 72 °C for 1 min. Final elongation was completed at 72 °C for 10 min. PCR products were set in 1.5% agarose gel, stained with ethidium bromide, and examined by electrophoresis at 100 V for 30 min. The gel was visualized using the Gel Doc XR þ system (Bio-Rad, Hercules, CA, USA). PCR products were sent to First BASE for sequencing.

### 2.5. Phylogenetic Analysis

The nucleotide sequences obtained from the respective primers, ITS1 and ITS4, were assembled in Chromas Pro 1.41 (Technelysium Pty Ltd., South Brisbane, Australia). The sequences were aligned with sequences retrieved from DNA databases (DDBJ and NCBI) using Bioedit.

The GenBank accession numbers, strain codes, and taxon names used in this study are presented in Table 1. Phylogenetic analysis was conducted using the neighbor-joining (ML) method implemented in MEGA 6.0. The strength of the internal branches of the phylogenetic trees was tested using bootstrap analysis with 1000 replications [27].

### 2.6. Antimicrobial Activity

The antimicrobial activity of tempeh extract against pathogenic bacteria was investigated. The pathogenic bacteria used for these experiments were *Escherichia coli* BCRC 10239, *Staphylococcus aureus BCRC* 10451, and *Bacillus cereus* BCRC 10927. Antimicrobial assays were performed using the paper disc method [28]. Bacteria were spread onto 15 mL soft agar containing 1.5% (*w*/*v*) agar. Then, a sterilized paper disc of 8 mm in diameter was placed on the agar plate, and 50 μL of tempeh extract was added, with ampicillin (0339 Amresco, Solon, OH, USA) as a standard followed by Upadhyay [29], as it is the penicillin group of beta-lactam antibiotics and is able to penetrate Gram-positive and some Gram-negative bacteria. Plates were then incubated at 37 °C for 24 h. Finally, the antimicrobial activity was analyzed by observing the clear zones around the paper discs containing tempeh extract; these clear zones were regarded as inhibitory zones, and were recorded in mm.

### 2.7. γ-Aminobutyric Acid Determination

Samples were prepared according to a previously-described method with some modifications [25]. A total of 20 μL of the supernatant were then blown dry using nitrogen, and then a mixture of 40 μL ethanol:water:triethylamine (2:1:1 of volume ratio) was added for derivation, and then the mixture was blown dry with nitrogen (flow rate 1.5~2 L/min, nitrogen pressure: 0.03 Mpa). Then, a mixture of 60 μL ethanol:water:triethylamine:ethyl isothioate (7:1:1:1 of volume ratio) was added and allowed to react for 20 min. It was then dried with nitrogen. A total of 20 μL of the mobile phase was dissolved, filtered through a 0.45 μm filter (Sartorius AG, Göttingen, Germany), and analyzed using a Hitachi Chromaster HPLC system (Hitachi Co., Tokyo, Japan), which included a chromatographic eluent 5160 pump, 5260 autosampler, and 5430 Diode Array Detector (DAD). HPLC column: reverse-phase C18 column (25 cm / 4.6 mm inner diameter, 5 μm, Kanto Chemical. CO., Inc., Tokyo, Japan). The HPLC conditions were as follows: mobile phase A:B = 80:20; A: 8.205 g of nitric acid (CH_3_COONa), 0.5 mL of triethylamine, 0.7 mL of acetic acid, and 5 mL of acetonitrile, pH 5.8; B: acetonitrile: water = 60:40, pH 5.8; flow rate: 0.6 mL/min; absorbance: 254 nm).

### 2.8. Animal Study

Sixty mixed-sex adult zebrafish (Ethic approval code and date: MU-2020-0103, 108/12/28) of the *Danio rerio* strain were held in tanks with constant aeration equipped with biological filtering under a natural photoperiod (approximately 14 h light: 10 h dark). The water was maintained at 25 ± 2 °C and pH 7.5 ± 0.25. The zebrafish were divided into five tanks of 30 × 20 × 20 cm, twelve fish per tank, acclimatized for a week, and fed with commercial flake food (Hikari Co, Ltd., Matsuyama-shi, Ehime, Japan). Twenty-four hours later, the test substance of red-bean tempeh extract was added to the water to expose the fish for 15 min. The fish were then submitted to a stress stimulus, which involved chasing the fish with a net for two minutes. The fish were then sampled after 0, 60, 120 and 240 min for whole-body cortisol analysis. There were five tanks one for each group: in the control group, fish were only chased with a net for two minutes; in the ethanol group, 75%, 50 mL ethanol added to the take (final concentration 0.75% alcohol) and in three tempeh groups, fish were treated with 30 mL of one of three tempeh extract solutions for 15 min. The tempehs were created by fermentation of red beans with three fungal strains: (1) *R. microsporus* var. *oligosporus* IFO 8631 RO (ATCC 22959); (2) co-culture with RO and *Lactobacillus rhamnosus* BCRC16000; and (3) our screened Strain 1. The experiments were performed in triplicate. To measure whole-body cortisol, we examined one fish from each strain (approximately 0.5 g of tissue) at each of the four time points totaling twelve fish. For cortisol extraction, fish were captured and immediately frozen with liquid nitrogen for 10–30 s and stored at −80 °C until cortisol extraction. Cortisol extraction on whole-body samples procedures were using the method described by Alderman and Bernier [30]. Briefly, body samples were partially thawed, weighed, and then homogenized in 500 µL of ice-cold 10 mM phosphate-buffered saline (PBS). After recording the weight (g), the whole-body sample was dissected on ice into smaller parts for efficient homogenization. Homogenization was performed using a mortar and pestle, following which the homogenate was collected in a 15 mL centrifuge tube. The mortar and pestle were washed with 75% ethanol and de-ionized water in between homogenizations of samples, an important step to minimize the cross-contamination of samples. A total of 5 mL of ethyl ether were added to each sample tube, the tubes were vortexed for 1 min, and then centrifuged at 3500 rpm for 5 min by centrifuge (GmbH—Z 323 K, Hermle LaborTechnik, Wehingen, Germany). Following centrifugation, the organic layer containing cortisol was removed from each sample and placed in a separate test tube. The process was repeated 2 (or 3) times consistently throughout the experiment to ensure maximal cortisol extraction. The cortisol-containing layer was usually yellowish in color. The samples were evaporated under nitrogen purging and subjected to a cortisol ELISA assay. The analysis was performed using a cortisol ELISA kit (Cortisol ELISA 96 Well kit, ADI-900–071, Enzo Life Sciences, New York, US), and all reagents were maintained at room temperature for at least 30 min prior to opening. All standards and samples were run in triplicate, and the optical density was read at 405 nm, preferably with correction between 570 and 590 nm. The data were managed using an immunoassay software package with a 4-parameter logistic (4PL) curve-fitting program.

### 2.9. Statistical Analysis

The data were analyzed by one-way analysis of variance (ANOVA) using SigmaStat software (Systat Software Inc., Point Richmond, CA, USA). *p*-values < 0.05 were considered significant.

## 3. Results

### 3.1. PCR, DNA Electrophoresis and Phylogenetic Analysis

Five strains were cultured for a day. Their mycelia was used as the DNA source. Amplification of the ITS rDNA region was performed using PCR with the primer pair ITS1 (5′-TCCGTAGGTGAACCTGCGG-3′) and ITS4 (5′-TCCTCCGCTTATTGATATGC-3′). The results of DNA electrophoresis are shown in Figure 1. *R. microsporus* var. *oligosporus* IFO 8631 (ATCC 22959) and Strains 1, 2 and 3 had the same 18S rRNA sequence of approximately 600 bp, and Strain 4 was approximately 800 bp. The results show that Strains 1, 2, 3 and 4 had rRNA sequences ranging from 0.5 to 1.0 kb; the sequences of strains RO, 1, 2 and 3 were around 0.5 kb, while Strain 4 was around 1.0 kb.

Based on the ITS tree (Figure 2) generated using neighbor-joining analysis, the Mucorales strains we used were divided into two groups, Genus *Rhizopus* and Genus *Lichtheimia*. Strains *R. oryzae*, *R. azygosporus*, *R. stolonifera, L. hyalospora*, *L. sphaerocysis, L. corymbifera, L. omata* and *L. ramosa* were also classified into specific taxa, respectively. Among the strains in this study, Isolate 1, Isolate 2, Isolate 3 and all *R*. *oryzae* strains were assigned to the same clade, the three isolates belonging to *R. oryzae*. Isolate 4 was found in the branch of the clade of *L. ramosa* in the group of *Lichtheimia*, indicating that Isolate 4 was a member of *L. ramose*.

### 3.2. Antibacterial Activity

This study evaluated the antibacterial properties of four strains of fungus isolated from red-bean tempeh and an inoculum with RO as the standard. Three bacteria known to affect human health were used. Using the paper disc method, we found antibacterial effects against *S. aureus* and *B. cereus* but none against *E. coli*. Clear zones of all of strains were observed and measured: + represented a clear zone equal to 5 mm; ++ a clear zone of 5 to 10 mm; and +++ a clear zone larger than 10 mm; while – indicated no antibacterial activity. Ampicillin was used as the antibacterial agent. It had the greatest antimicrobial activity (*E. coli* BCRC10239 was ++; *S. aureus* BCRC10451 was ++++, and *B. cereus* ATCC10361 was +++). Strain 2 exhibited the best antibacterial activity with *B. cereus*, with a clear zone of 8 mm (between 5 and 10 mm, ++), while the other strains had clear zones of + (5 mm) or – (no clear zone) (Table 1).

### 3.3. GABA Detection

Six concentrations (0, 200, 400, 600, 800, 1000 g kg^−1^) were used to optimize the HPLC chromatographic conditions. The contents of GABA in different red-bean tempeh samples by using six different strains are shown in Table 2. Red-bean tempeh fermented by co-culture of *R. microsporus* var. *oligosporus* IFO 8631 (ATCC 22959) and *Lactobacillus rhamnosus* BCRC16000 had the highest GABA content (19.0 ± 1.8 g kg^−1^) and Strain 1 had the second highest content (10.8 ± 2.1 g kg^−1^). The third highest GABA content was observed for standard strain *R. microsporus* var. *oligosporus* IFO 8631 (ATCC 22959) (10.4 ± 5.2 g kg^−1^). The lowest was found in Strain 4 (7.69 ± 0.2 g kg^−1^). In red-bean tempeh fermented with Strains 2 and 3, no GABA was detected (Table 2).

### 3.4. Animal Study

The zebrafish were exposed to 50 mL of the three tempeh extract solutions (RO, tempeh fermented by *R. microsporus var. oligosporus* RO (ATCC 22959); co-culture, co-culturing *R. microsporus* var. *oligosporus* IFO 8631 (ATCC 22959) and *Lactobacillus rhamnosus* BCRC16000; and Strain 1) as well as ethanol only (75% alcohol, 50 mL) and no treatment at all as controls for 15 min. They were then submitted to acute stress (Figure 3). Cortisol reduction was assessed post stress at 0, 60, 120, and 240 min. Exposure to tempeh fermented with Strain 1 had the greatest effect on the whole-body cortisol level, resulting in an impaired cortisol response to the acute stressor at 240 min (756.7 ± 71.5 ng g^−1^) as compared with the RO group (4239.4 ± 338.6 ng g^−1^), though the fish exposed to Strain 1 and stress had lower pre-stress cortisol concentrations than the RO tempeh + stress groups.

## 4. Discussion

In this study, the first to substitute one bean, a small red bean grown in Taiwan, for soybean to produce a traditional fermented food stuff known as tempeh and assess its possible health effects. All strains had antibacterial activity on *S. aureus* and *B. cereus*, particularly Strain 2 (Table 1). We measured GABA content, antibacterial activity and cortisol level of tempeh fermented with several strains of *Rhizopus* including those screened in banana leaves, a purchased strain, and co-cultured group. We found the highest level GABA in the co-culture group (19.028 ± 1.831 g kg^−1^), followed by screened Strain 1 (10.57 ± 2.14), the purchased strain (10.440 ± 5.282), and screened Strain 4 (7.69 ± 0.272) (Table 2). The extracts, especially the one made with Strain 1 tempeh, significantly reduced cortisol in zebrafish, the longer the exposure, the greater the reduction (*p* < 0.05) (Figure 3).

Tempeh is traditionally made with soybean and has been made with barley [6]. Some agricultural areas in southern Taiwan produce a special small red bean whose flavor and aroma the Taiwanese particularly appreciate. Xiao et al. [31] reported that the fermentation of red bean flour with *Cordyceps militaris* (L.) Fr. improved the flour’s nutritional, physicochemical, and functional properties and increased its biological activity. Chou et al. [32] compared the ability of 50% ethanol extract of red beans fermented and non-fermented with *Bacillus subtilis* to improve antioxidant status in aged Institute of Cancer Research (ICR) mice and found that the fermented red bean extract greatly improved such antioxidant properties as vitamin E level and superoxide dismutase activity. Therefore, we wanted to investigate whether fermented red bean could be used innovatively as a nutritional and functional ingredient in food design for this study, the traditional Indonesian soybean-made tempeh.

We found red-bean tempeh fermented using a co-culture of *R. microsporus* var. *oligosporus* IFO 8631 (ATCC 22959) and *Lactobacillus rhamnosus* BCRC16000 had the highest GABA content (19.028 ± 1.831 g kg^−1^), Strain 1 tempeh had the second highest (10.6 ± 2.1 g kg^−1^), followed by standard strain *R. microsporus* var. *oligosporus* IFO 8631 (ATCC 22959), the third highest (10.4 ± 5.2 g kg^−1^), and Strain 4, the lowest GABA content (7.69 ± 0.2 g kg^−1^). The other two strains produced no GABA. Our co-culture group had the highest GABA content. Li et al. [33] also studied the effect of co-culture of *Lactobacillus brevis* GABA and *Aspergillus oryzar* CACC 40250 in the production of GABA in doenjang, a type of soybean-based sauce, and found that it significantly increased the production rate of GABA as well as a couple of isoflavones.

We used paper disc diffusion to assess the antibacterial activity of tempeh fermented with our four screened fungus strains, a purchased strain with proven stability, our co-cultured tempeh, as well as ampicillin-treated extract (control) on *E. coli*, an indicator of freshness and spoilage, and *S. aureus* and *B. cereus*, two bacteria often associated with food poisoning. The extract containing ampicillin had the greatest antibacterial activity. Extract from Strain 2 had a strong antibacterial effect on *B. cererus*, while the others had little or no effect. Lee et al. [28], using paper disk diffusion as well as viable cell counting to investigate the effect of antimicrobial effect of extracts made from the flower *Inula Britannica* against several *Helicobacter pylori*, a common cause of gastric ulcers, found that *I. britannica* extracts could significantly reduce cell counts of three strains of *H. pylori* in vitro and concluded that such an extract could be used to produce novel functional foods. Ross et al. [12] also used paper disc diffusion to study the effect of co-culturing *Rhizopus* with *Burkholderia gladioli pv cocovenenans*. Although they were studying the harmful effects of the co-cultured pair, the resulting images of their study clearly illustrated the production of bongkrekic acid, which is poisonous to humans. It was found in their study to be able to block both bacterial and fungus growth, which was clearly observed in their images. Their study highlights the context-dependent formation of antifungal and antibacterial agents at the fungal–bacterial interface.

In this study, we measured GABA content in tempeh extract. GABA has many functions and has been found to have antihypertensive, anti-insomnia, as well as anti-anxiety effects. Cortisol levels have also been associated with stress levels in humans. One study assessed the effect of two anti-anxiety drugs on cortisol levels in zebrafish, and found that the two psychoactive drugs decreased cortisol levels in these fish [23]. In this study, we measured the effect of our red-bean tempeh extracts on pre-stress and post-stress cortisol levels in zebrafish and found the extract from Strain 1 produce the strongest reduction of post-stress cortisol in these fish at 240 min (756.7 ± 71.5 ng g^−1^), followed by the RO group (4239.4 ± 338.6 ng g^−1^) and co-culture group (3981.0 ± 1109.8 ng g^−1^). All those exposed to the extracts but not those exposed to alcohol only had significantly lower cortisol levels than the controls (*p* < 0.05). However, those exposed to the extract of Strain 1 were found to have significantly lower post-stress cortisol levels than the other two extracts studied (*p* < 0.05). The fish subjected to Strain 1 had lower pre-stress cortisol concentration than those subjected to the other extracts, possibly because Strain 1 had higher GABA levels. Note that fish exposed to alcohol only did not have significantly lower levels of post-stress cortisol at 240 min than the control group (3989.6 ± 308.5 ng g^−1^ vs. 4010.3 ± 139.7 ng g^−1^, respectively (Figure 3). Therefore, the use of alcohol as solvent did not confound our results.

It was interesting to note that although the red-bean tempeh fermented by co-culture had the highest GABA content (19.0 ± 1.8 g kg^−1^), the zebra fish exposed to it did not have the lowest post-stress cortisol. The fish exposed to Strain 1 had the greatest reduction in cortisol levels and best inhibition of cortisol response to an acute stressor at 240 min (756.7 ± 71.5 ng g^−1^), though it had the second highest GABA levels (10.5 ± 2.1 g kg^−1^). Fish exposed to extracts with the highest GABA contents did not have a significantly better reduction in cortisol levels than Strain 1, suggesting that the GABA in these extracts was not the only reason that cortisol (stress response) was reduced in the zebra fish. Ahmad et al. [34] compared the neuroprotective effects of total isoflavones obtained from soybean and tempeh against scopolamine-induced cognitive dysfunction and found that other molecules, such as isoflavone, may also play important role in anti-stress effect. Therefore, the extracts we studied may have other components not studied here that affect stress response.

This study has several limitations. One limitation is that we only measured GABA but not isoflavones, which may also affect stress responses. Another limitation is that we only used cortisol level to measure stress response. Other studies, such as animal behavior, could possibly be used. Still, another limitation is that we tested antimicrobial activity against three strains of bacteria only. More strains could be used in the future.

## 5. Conclusions

Based on the results of this study, our *Rhizopus*-fermented red-bean tempeh was found to be nutritious, as evidenced by GABA content, which is beneficial due to its effect on cortisol level under pre-stress and post-stress conditions. Considering the Taiwanese appreciation of the red bean, there may be viable market for red-bean tempeh and it may be consumed with similar health benefits of fermented soybean and barley tempeh, though further studies in human subjects may be needed to make this conclusion. In addition, further study is needed to analyze other nutritional components of red-bean tempeh with other health indicators.

## Figures and Tables

**Figure 1 microorganisms-08-01330-f001:**
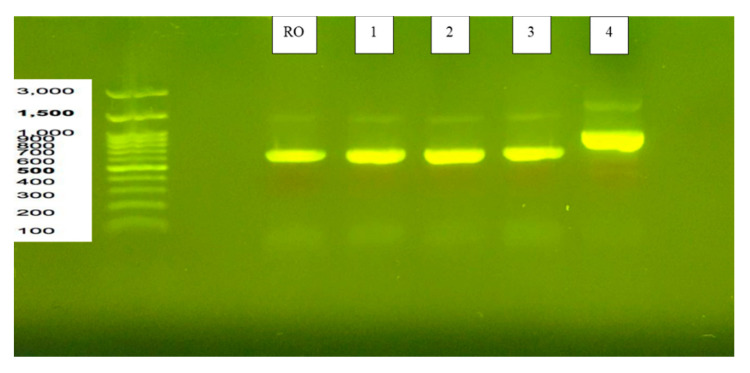
Results of agarose gel electrophoresis analysis for DNA identification of 4 strains isolated from natural tempeh. Strains 1, 2, 3 and 4 had rRNA sequences ranging from 0.5 to 1.0 kb; strains RO, 1, 2 and 3 had the same sequence of around 0.5 kb, while Strain 4 was around 1.0 kb.

**Figure 2 microorganisms-08-01330-f002:**
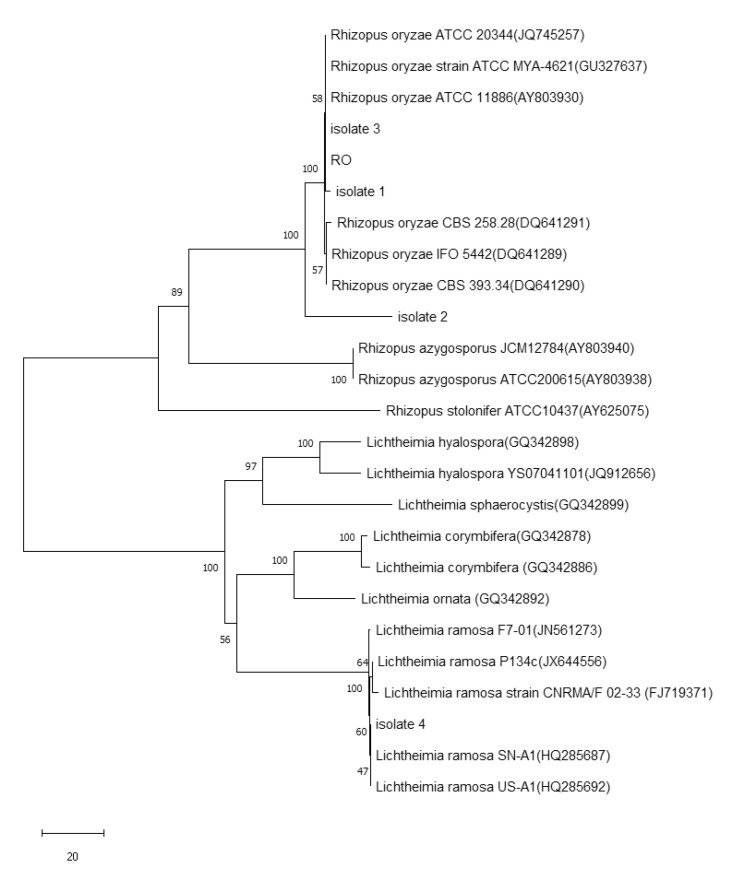
Phylogenetic tree obtained using the neighbor-joining method for 18S rDNA sequences of strains 1, 2, 3 and 4, and members of related fungi. The numbers at the branch points are bootstrap values for parsimony-based analysis.

**Figure 3 microorganisms-08-01330-f003:**
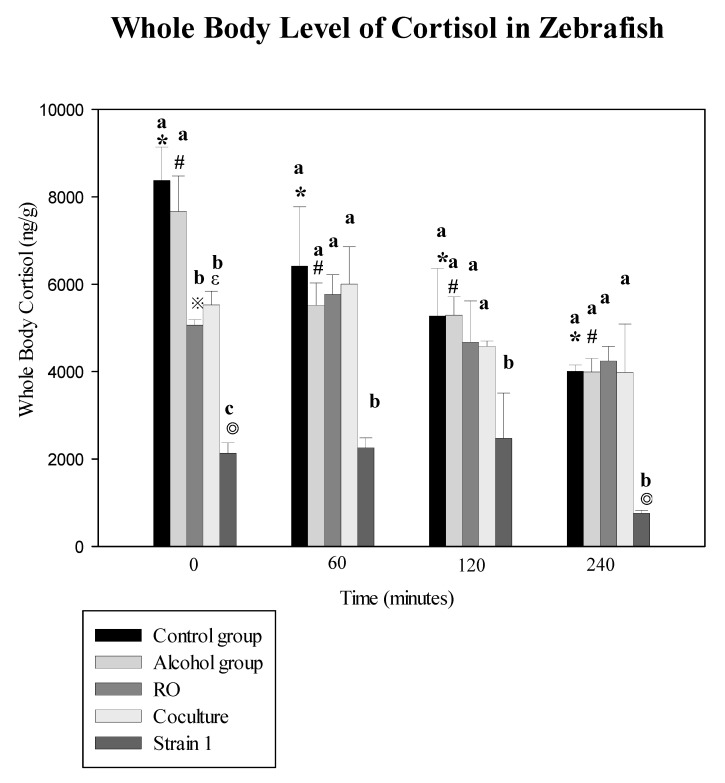
Kinetics of whole-body cortisol secretion in zebrafish subjected to treatment with tempeh ferments. The vertical axis represents the whole-body cortisol secretion of the zebrafish (ng/g). Bars show means of triplicates ± SD. *, #, ※, ε, and ◎ represent *p* < 0.05 as compared with the 0-min group cortisol level (ng g^−1^). a–c: the same bar with different letters are significantly different, *p* < 0.05.

**Table 1 microorganisms-08-01330-t001:** Antimicrobial activity of tempeh substances produced by four kinds of fungus (1,2,3, and 4), *R. microsporus* var. *oligosporus* (RO) (*R. microsporus* var. *oligosporus* IFO 8631 RO (ATCC 22959), and coculture (red bean inoculated with RO and *Lactobacillus rhamnosus* BCRC 16000).

Indicator Strains	Ampicillin	Strain 1	Strain 2	Strain 3	Strain 4	RO Red Bean	RO Co-Culture
*E. coli* BCRC 10239	++	−	−	−	−	+	+
*S. aureus* BCRC 10451	++++	+	+	+	+	+	+
*B. cereus* BCRC10927	+++	+	++	+	+	+	+

+ represents a clear zone equal to 5 mm; ++ a clear zone of 5 mm to 10 mm; +++ a clear zone of 10 mm to 15 mm; ++++ a clear zone larger than 15 mm; and − no antibacterial activity. Ampicillin was used as the antibacterial agent.

**Table 2 microorganisms-08-01330-t002:** Contents of γ-aminobutyric acid (GABA) detected by HPLC in different red-bean tempeh samples by using 6 different strains.

Sample	Concentration (g kg^−1^)
RO	10.440 ± 5.282
RO+LAB	19.028 ± 1.831
Strain 1	10.57 ± 2.154
Strain 2	UD
Strain 3	UD
Strain 4	7.69 ± 0.272

UD = undetected, and the values show means of triplicates ± SD.

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
