# Peer review of "Effects of Red-Bean Tempeh with Various Strains of Rhizopus on GABA Content and Cortisol Level in Zebrafish"

_microorganisms, 2020, doi:10.3390/microorganisms8091330_

Round 1

Reviewer 1 Report

Section 2.1 

Malt extract powder: Please indicate brand, city, and country

RO water: please spell out RO at the first time, then you can use acronym later in the text

Slant medium: give more detailed information on the recipe or brand, city and country

Sterlized gauze: list the brand, city, country. 

MRS broth: If it is self prepared, please provide the recipe and reference. If purchased, please give the brand and company information

..sterilized for 15 minuttes (at 121C 15 psi)...: Please indicate what equipment do you use. 

Degraded mineral oil: What is that? If it is purchased, indicate brand and company information. 

0.5 ml was taken before the experiment...: Please specify when this 0.5 ml samples were taken. 

Section 2.2

"Red bean were Soaked in lactic acid of pH 3.5...": please indicate that % of lactic acid, and how many g of bean into how much lactic acid. 

"1g of fermented red beans was extracted using 15 ml ...." Change to "One gram of ....."

"... 15ml of 70% alcohol and shook 1 minutes,.." change to "shook for 1 minutes..."

Section 2.3

" 1 g of banana leaf was mixed...and vortexed." change to " one gram of banana....", and Vortexed for how long.

"...MEA+ampicillin": If it is self prepared, please provide the recipe and reference. If purchased, please give the brand and company information

Section 2.4

"Phytopure TM DNA extraction Kit", please indicate company/brand name, model no. city, and country.

Section 2.6

"... were regarded as inhibitory zones, and were recorded in mm". Please specify more clear how do you measure the zone in mm, by diameter? It would be more helpful if you can have diagram to show how this is measured. 

Section 2.7

"... ethanol:water:triethylamine (2:1:1) " and " ethanol:water:triethylamine:ethyl isothioate (7:1:1:1)" indicate if it is volume ratio or weight ratio. 

"... then blown dry with nitrogen." How to blow dry with nitrogen?

"... dried with nitrogen." How to dried with nitrogen, please discribe more details of the method. 

"...0.45 um filter". Brand, material of the filter, and company information. 

HPLC: what column is used. 

Section 2.8

"... commercial flake food" if it is purchased, please indicate the brand and company information. 

"... fish were exposed to the test substance..." what test substance are you refering. 

"These experiments were replicated three times" Change to "...were triplicated"

"Homogenized in 500ul of ice-cold 1X phosphate-buffred saline (PBS)" Please indicate the concentration or composition of the PBS. 

"...tubes were vortexed for 1min then centrifuged at 3500 rpm for 5min" Brand, model and company of the centrifuge. 

Please add a Conclusion section to the end. 

"

Reviewer 2 Report

The manuscript "Biological activities of red bean tempeh fermented by screened Rhizopus species" is of interest.

However, there are some significant shortcoming that I believe need to be addressed before making a decision regarding acceptance of the manuscript.

The first point is the fact that it is a poorly written paper. The English language needs to be significantly improved. It is much more than spelling and grammar, it is also structure of paragraphs and the flow of the 'story line' within the paper. 

The abstract is very confusing/disorganised. The authors are trying to convey one message at the time (one per sentence), however there is no connectivity between the sentences. Don't make the reader guess what you are trying to say. Use more words and linker sentences. There is no real word limit to the manuscripts for this journal.

Double check you punctuation. The abstract sentence that starts with "Animal study of zebrafish ...", finished with a comma and a full-stop. This is easy to fix and should have been picked up in the proof reading before submission. Also that start of the sentence is poor English.

in the 2nd sentence of the introduction the authors claim that phytic acid and tannins etc are responsible for flatulence. This is not true. The following sentence is correct.

Nowhere in the manuscript do the authors provide a reference to the production or existence of red bean tempeh. There is an explanation of soy-based tempeh, but nothing of red bean tempeh.

In the methodology, first sentence - remove the words 'fungus strains' and 'bacteria'. The readers will know what the organisms are by their names.

Please write all names of organisms in italics. There are a couple of occurrence throughout the MS where this did not happen. 

At 2.2, it is clearer if you write: "Red beans were soaked in a lactic acid acidified aqueous solution at pH 3.5 and boiled for 30 minutes." Please indicate how long they were soaked for.

Don't start sentences with a number - write the numbers in full. I.e. One hundred instead of '100' and One instead of '1'. there are multiple occurrences of this in the methodology, while it is written correctly in the rest of the manuscript.

At 2.5, move the reference for 'bioedit' to the reference section.

It strikes me that the data in table 1 is essentially the same as that presented in figure 2. Please do not present the same data twice.

There is no explanation to why ampicillin was used as the control antibiotic. Why this one, and not some other or multiple other antibiotics?

Also, there is no explanation to why a co-culture was used for the RO experiments, but not for the isolated strains. Please explain the relevance of this.

When it comes to the animals study, at one point it is stated that 48 zebreafishes were divided into four tanks, but further down you are referring to five tanks.  Which one is it?

You state that the fish were exposed to the test substance for 15 minutes. How did you do that? I assume that you added the substance to the tank, but how did you remove it again from the tank? Please make sure that your methodology is explained with enough detail so that someone else can repeat your work.

Results

first sentence - don't say 'for a day'. That is very unscientific. Say "for 24 hrs".

The second and third sentence should be removed. All you do is to repeat stuff that is already explained in the methodology.

at 3.2, Remove the second sentence. You have already explain which bacteria you are using in the methodology.

Further down, you write "Ampicillin was used as the ...", add the word 'control' between the and antibacterial. Please keep in mind that you will have to explain why you used ampicillin.

Discussion.

The discussion has no structure. You are jumping from one half-fact to the next half-fact. It is nearly impossible to follow your discussion.

At the bottom of pg 9, you write "... the ethanol solvent effect has low possibility to affect ...". I get what you are trying to say, but it is poor English.

Later in the discussion, you are starting to include abbreviations (RBE, RBNE, and HPA) that are never used again. Only use abbreviations when they are used at least 3 times, otherwise delete them.

Page 10 (starting with Xiao et al [29]) is a very random collection of statements that don't really provide a clear story. In fact, you don't actually point out how the various statements you make relate to your study. You state that "We examined several different beans and cereals ...". That is not true at all, you only examined red beans.

Further to mt comment above - I am curious to find out what the relevance is of the sentence starting with: "Chibucos et al [32] released ...". I have no idea how that related to your findings. The readers of this manuscript are not mind-readers. You have to be explicit.

At some point in time you will have to fix up your references. At times the titles are written with capital letters and at other times they are not.

Author Response

Review 2

Comments and Suggestions for Authors

The manuscript "Biological activities of red bean tempeh fermented by screened Rhizopus species" is of interest.However, there are some significant shortcoming that I believe need to be addressed before making a decision regarding acceptance of the manuscript.

1. The first point is the fact that it is a poorly written paper. The English language needs to be significantly improved. It is much more than spelling and grammar, it is also structure of paragraphs and the flow of the 'story line' within the paper.

ANS: We have found a skilled writer to help us to improve English writing.

2. The abstract is very confusing/disorganised. The authors are trying to convey one message at the time (one per sentence), however there is no connectivity between the sentences. Don't make the reader guess what you are trying to say. Use more words and linker sentences. There is no real word limit to the manuscripts for this journal.

ANS: We have find a skilled writer to help us to revise the abstract.

3. Double check you punctuation. The abstract sentence that starts with "Animal study of zebrafish ...", finished with a comma and a full-stop. This is easy to fix and should have been picked up in the proof reading before submission. Also that start of the sentence is poor English.

ANS: We have found a skilled writer to help us to improve English writing and double check our punctuation.

4. in the 2nd sentence of the introduction the authors claim that phytic acid and tannins etc are responsible for flatulence. This is not true. The following sentence is correct.

ANS: We have delete the sentence and reference.

5. Nowhere in the manuscript do the authors provide a reference to the production or existence of red bean tempeh. There is an explanation of soy-based tempeh, but nothing of red bean tempeh.

ANS: It was common for soybean tempeh article but hardly found about red bean tempeh article. We have mentioned about the merits of red bean in reference [1], [2]

6. In the methodology, first sentence - remove the words 'fungus strains' and 'bacteria'. The readers will know what the organisms are by their names.

ANS: We have removed the words 'fungus strains' and 'bacteria'

7. Please write all names of organisms in italics. There are a couple of occurrence throughout the MS where this did not happen.

ANS: We have checked about names of organisms in italics

8. At 2.2, it is clearer if you write: "Red beans were soaked in a lactic acid acidified aqueous solution at pH 3.5 and boiled for 30 minutes." Please indicate how long they were soaked for.

ANS: We have changed to “Red beans were soaked in 1% lactic acid, and followed by cooking at 100 °C for 30 minutes”.

9. Don't start sentences with a number - write the numbers in full. I.e. One hundred instead of '100' and One instead of '1'. there are multiple occurrences of this in the methodology, while it is written correctly in the rest of the manuscript.

ANS: We have corrected the numbers to full number name.

10. At 2.5, move the reference for 'bioedit' to the reference section.

ANS: We have deleted 'bioedit' to the reference section.

11. It strikes me that the data in table 1 is essentially the same as that presented in figure 2. Please do not present the same data twice.

ANS: We have deleted original Fig 3 as it is the same result as table 1.

12. There is no explanation to why ampicillin was used as the control antibiotic. Why this one, and not some other or multiple other antibiotics?

ANS: We have cited Upadhyay [28], in their paper they used ampicillin as a control group for pineapple stem waste antimicrobial experiment. Besides, ampicillin is in the penicillin group of beta-lactam antibiotics, and is able to penetrate Gram-positive and some Gram-negative bacteria.

13. Also, there is no explanation to why a co-culture was used for the RO experiments, but not for the isolated strains. Please explain the relevance of this.

ANS: RO (R. microsporus var. oligosporus IFO 8631 RO (ATCC 22959)) is the standard rhziopus strain purchased from Collection Research Center (BCRC), Food Industry Research and Development Institute (Hsinchu, Taiwan). It is more stable and reliable for co-culture experiment. We will use strain 1 we screened next time.

14. When it comes to the animals study, at one point it is stated that 48 zebrafishes were divided into four tanks, but further down you are referring to five tanks. Which one is it?

ANS: We have corrected to sixty zebrafishes, and divided to five tanks.

15. You state that the fish were exposed to the test substance for 15 minutes. How did you do that? I assume that you added the substance to the tank, but how did you remove it again from the tank? Please make sure that your methodology is explained with enough detail so that someone else can repeat your work.

ANS: We have mentioned in sentence 236 to 243:

Twenty-four hours later, the test substance of red bean tempeh extract was added to the water to expose the fish for 15 minutes. The fish were then submitted to a stress stimulus which involved chasing the fish with a net for two minutes. The fish were then sampled after 0, 60, 120 and 240 minutes for whole-body cortisol analysis. There were five tanks one for each group: in the control group, fish were only chased with a net for two minutes; in the ethanol group, 75%, 50mL ethanol added to the take (final concentration 0.75% alcohol) and in three tempeh groups, fish were treated with 30 ml of one of three tempeh extract solutions for 15 minutes.

Results

16. first sentence - don't say 'for a day'. That is very unscientific. Say "for 24 hrs".

ANS: We have corrected to "for 24 hrs".

17. The second and third sentence should be removed. All you do is to repeat stuff that is already explained in the methodology.

ANS: We have removed the sentences.

18. at 3.2, Remove the second sentence. You have already explain which bacteria you are using in the methodology.

ANS: We have removed the sentence.

19. Further down, you write "Ampicillin was used as the ...", add the word 'control' between the and antibacterial. Please keep in mind that you will have to explain why you used ampicillin.

ANS: We have cited Upadhyay [28], in their paper they used ampicillin as a control group for pineapple stem waste antimicrobial experiment. Besides, ampicillin is in the penicillin group of beta-lactam antibiotics, and is able to penetrate Gram-positive and some Gram-negative bacteria.

Discussion.

20. The discussion has no structure. You are jumping from one half-fact to the

  1. next half-fact. It is nearly impossible to follow your discussion.

ANS: We have found a skilled writer to help us to rewrite the Discussion.

21. At the bottom of pg 9, you write "... the ethanol solvent effect has low possibility to affect ...". I get what you are trying to say, but it is poor English.

ANS: We have changed the sentence to “Therefore, the use of alcohol as solvent did not confound our results.” As in the revised manuscript sentence 386 ~ 387.

22. Later in the discussion, you are starting to include abbreviations (RBE, RBNE, and HPA) that are never used again. Only use abbreviations when they are used at least 3 times, otherwise delete them.

ANS: We have deleted them.

23. Page 10 (starting with Xiao et al [29]) is a very random collection of statements that don't really provide a clear story. In fact, you don't actually point out how the various statements you make relate to your study. You state that "We examined several different beans and cereals ...". That is not true at all, you only examined red beans.

ANS: We have corrected it.

24. Further to mt comment above - I am curious to find out what the relevance is of the sentence starting with: "Chibucos et al [32] released ...". I have no idea how that related to your findings. The readers of this manuscript are not mind-readers. You have to be explicit.

ANS: We have deleted the reference.

25. At some point in time you will have to fix up your references. At times the titles are written with capital letters and at other times they are not.

ANS: We have fixed up our references.

Round 2

Reviewer 2 Report

It is clear that the English language editor did a great job improving the readability of this manuscript.

In combination with addressing the other reviewer's comments, this manuscript to ready for acceptance. However, I still have one outstanding issue I would like the authors to address. In the first sentence of section 2.2. the authors state that the beans were soaked in a lactic acid solution prior to being cooked. Please indicate for how long the beans were soaked in that lactic acid solution. Without this information, others cannot repeat the experiments to get the same results.

Author Response

In combination with addressing the other reviewer's comments, this manuscript to ready for acceptance. However, I still have one outstanding issue I would like the authors to address. In the first sentence of section 2.2. the authors state that the beans were soaked in a lactic acid solution prior to being cooked. Please indicate for how long the beans were soaked in that lactic acid solution. Without this information, others cannot repeat the experiments to get the same results.

ANS: Original: “Red beans were soaked in 1% lactic acid, and followed by cooking at 100 °C for 30 minutes”

We changed the paragraph to “Red beans were washed and soaked for twelve hrs. After drying, water of twice the weight of red beans and 1% lactic acid were added, and followed by cooking at 100 °C for 30 minutes”

Red beans were soaked in 1% lactic acid and cooked immediately for reducing pH value and softening red bean, as its antiseptic effect and easier to cook